# Sensing and Forecasting Crowd Distribution in Smart Cities: Potentials and Approaches

Alket Cecaj [1,2,*], Marco Lippi [1,2], Marco Mamei [1,2] and Franco Zambonelli [1,2]

1 Dipartimento di Scienze e Metodi dell'Ingegneria, University of Modena and Reggio Emilia, 42122 Reggio Emilia, Italy; marco.lippi@unimore.it (M.L.); marco.mamei@unimore.it (M.M.); franco.zambonelli@unimore.it (F.Z.)
2 Artificial Intelligence Research and Innovation Center (AIRI), University of Modena and Reggio Emilia, 41125 Modena, Italy
* Correspondence: alket.cecaj@unimore.it

**Abstract:** The possibility of sensing and predicting the movements of crowds in modern cities is of fundamental importance for improving urban planning, urban mobility, urban safety, and tourism activities. However, it also introduces several challenges at the level of sensing technologies and data analysis. The objective of this survey is to overview: (i) the many potential application areas of crowd sensing and prediction; (ii) the technologies that can be exploited to sense crowd along with their potentials and limitations; (iii) the data analysis techniques that can be effectively used to forecast crowd distribution. Finally, the article tries to identify open and promising research challenges.

**Keywords:** crowd; forecasting; sensing; crowd-forecasting; predicting-methods; approaches

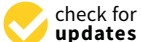



## 1. Introduction

A common general issue in the vast panorama of research and approaches in the context of smart cities is the need for sensing current situations [1] (e.g., about traffic, energy consumption, pollution, etc.) and the possibility of predicting the future evolution of such situations to prevent problems and to perform more informed decision making and planning.

The capacity to monitor and predict the behavior of crowds [2], in particular, is a fundamental enabling driver for smart cities. Without any need to know the behavior and movements of individual people (an issue that would also raise severe privacy concerns), having a picture of the overall density of people in the different parts of a city, understanding how such density changes over time, and possibly predicting such changes can be of great help in a large variety of situations: organizing events, managing situations of emergency, maintaining an efficient public transportation system, or even tracking how viruses spread through the urban areas.

The importance of crowd sensing (that is "sensing the crowd", not to be confused with "sensing by the crowd" approaches usually referred to as "crowdsensing" and concerning the possibility for people in the crowd to contribute sensorial information [3]) and crowd forecasting is being increasingly recognized, and a large number of approaches are being proposed. Nevertheless, a comprehensive overview of such a relevant theme is still missing from the literature, and filling this gap is the intended contribution of this paper. In particular, the paper will:

- Identify the many application areas in which crowd sensing and prediction can be effectively exploited, also reporting about relevant examples from the literature and from the real-world. These include: mobility planning, social planning, event management, crowd steering, and epidemiological planning.
- Overview the many sensing technologies that have the potential to be useful towards crowd sensing, and characterize the type of data one can obtain from them. These

technologies include mobile phones, Wi-Fi antennas, social networks, and surveillance cameras.

- Analyze the many forecasting methods that have been proposed in the literature, their extent of applicability, and their effectiveness. Indeed, those methods are based on different algorithmic approaches and cannot disregard the deployment of data generating technologies. Moreover, the specific application scenario together with the forecasting need for "how many hours/days ahead" we should forecast, may also be a determinant for the approach taken for the forecasting process.

Finally, the paper tries to identify some relevant open research challenges in the area of crowd sensing and mobility forecasting.

The remainder of this paper is organized as follows. In Section 2 we overview the many application areas of crowd sensing and forecasting. In Section 3 we present the possible data sources that can be used to model crowd distribution and dynamics. Section 4 describes techniques for crowd forecasting. Section 5 identifies some open challenges, and Section 6 concludes.

## 2. Potentials

The data collected from different technological devices and sensors have already shown their potential in modelling and representing a range of situations with direct applications in fields such as epidemic modeling, urban planning, and event management. In this section, we list the different potential application areas of crowd sensing and crowd forecasting, also by considering the type of data used (structured, unstructured or any combination of data) and the requirements about the forecasting scenario (real-time, short-term and/or long-term).

### 2.1. Mobility and Urban Planning

Human mobility is made of patterns which can be daily, weekly or based on any other time period also depending on the data domain. It is the presence of these kinds of patterns that keep repeating in time and space that makes mobility planning possible in the short and long term. In particular, the patterns establish a daily/weekly routine that allows the prediction of the mobility flows from the mobile trajectories in a given urban area and moment of the day.

There are methodologies for building the so-called Origin-Destination (OD) matrix which allows for reconstructing the mobility flows at a large scale. Yet, building the OD matrix from aggregated mobility data is much more challenging than doing it from single user trajectory data. However, there are some promising approaches, such as in [4,5] based on the identification of the most likely set of movements explaining changes in the crowd density. Finally, knowing what are the patterns of the crowd flows with respect to a given moment of the day can be helpful for predicting the travel demand: in [6], for example, the authors propose a location recommendation system that infers personal preferences while at the same time taking in consideration the constraints imposed by traffic and road capacity. A method for the evaluation of OD matrix estimation algorithms is given in [7]. It is worth emphasizing that this kind of analysis can be applied also to a specific portion of the population (e.g., tourists, elderly, minorities, immigrants) shedding light on a whole range of phenomena with socio-economic impact.

On a different perspective, two further examples of application in the domain of urban planning are given by occupancy prediction and parking availability prediction. Such tasks are crucial for any modern urban infrastructure management, and thus they have been studied by using many types of data and different prediction methods. Occupancy estimation can be performed from large mobile phone datasets [8] or other kind of sensor data [9], to build systems that can identify the geospatial distribution of occupants in order to track or predict their movements. Parking availability, aims to recommend to drivers parking locations which have the highest probability of providing at least one parking spot available at the estimated arrival time. Clearly, the availability of cheap IoT sensors has

made these problems easier to face. However, the lack of integration of the sensors data with the car navigators remains challenging.

### 2.2. Social and Economic Planning

Social planning is about ensuring that the efforts to deal with social and economic issues take place in a coordinated community-wide basis with long term goals. In a wider sense, social planning is the deliberate introduction of social values into economic and political processes [10]. Civic engagement and social capital as a form of collaborative planning have been extensively studied in [11,12].

Though social planning has been studied mainly through theoretical approaches, in recent years, large mobility data collections have shown to help in providing a better understanding. Mobility data can be especially useful when thinking of social planning as a time-phased programming and collaboration. The work in [13], for example, studies the synchronization aspects of mobile communication patterns (using CDR data, which will be described in Section 3) for expressing in a quantitative way the social capital and civic engagement of an urban area. Other works that use data and, in particular, social network data from Facebook to define and measure social capital are [14,15].

Another application field that can be framed within social planning is the analysis of migration patterns, both between and within a country. Such an activity is very important to devise effective policies for supporting management and integration of the immigrant population and of minorities in general. A recent research challenge has been proposed to analyze mobility and behavioral patterns of Syrian refugees in Turkey with the goal of identifying policies for better integration of the refugees population in the hosting country [16]. In this context, mobility patterns have been studied to understand mainly: *(i)* segregation and differences in mobility patterns between refugees and natives. *(ii)* workforce displacements and the hiring of refugees in undeclared work, *(iii)* schooling access and mobility for the refugees population. In all these areas the study of mobility patterns as enabled by current technologies (see Section 3) provides and important asset from data-driven policy making [16].

Similarly, mobility data have been also used as a proxy for economic development. On the one hand, patterns of movements and especially their variability is found to be well-correlated with economic development of an area [17]. On the other hand, mobility patters well describe and reflect the economic impact following major layoff [18].

### 2.3. Event Management

Managing physical events which are attended by thousands of people poses several challenges of organizational and technological nature. Depending on the situation, an event might be a concert in the main square of a city, but it might also be a protest/riot of a relatively large group of people, or a situation of emergency such as an earthquake (disaster management) forcing people to leave the buildings and gather in specific areas. In other cases, public administrations need to detect gatherings of people for public order and observance-of-the-law monitoring purposes.

However, in all these cases, the main task for a possible monitoring system is the automatic detection of the event and mobility data is one of the best sources to feed these kinds of systems [19–21]. Moreover, depending on the specific application, we might be interested in the automatic description of an event, i.e., by asking the question "Why are people gathering in there?". Social network data can be combined with mobility data for this kind of application, as described in [22].

In recent years, we have seen another data type that has shown to be very effective in helping researchers spot and eventually predict crowd dynamics: images and videos. Once analyzed with image processing and deep learning networks such as the recurrent and convolutional ones, images and videos have been successful in predicting crowd density [23], crowd flow [24] and crowd scene understanding [25].

## 2.4. Crowd Steering

An important aspect of predicting crowded events is ensuring crowd safety, so as to be able to steer the movements of the crowd before dangerous situations occur [26,27]. For example, consider those situations–also called stampedes–in which crowds of people while in movement, become very dense in proximity of obstacles or bottle-necks. These situations have shown to be very dangerous [28], and history is full of them: just to mention a few examples, consider the 2010 Love Parade music festival in Duisburg, Germany, or the celebration of the 2014 New Year's Eve in Shanghai, China, where 18 and 36 people lost their lives, respectively.

Crowd steering to prevent dangerous situations and support evacuation plans [29] can be enforced via several techniques (digital signage, voice announcements, or human "steerers" in the crowd) and strategies [30]. However, we emphasize that the same techniques can be adopted in urban environments or crowded buildings simply to prevent uncomfortable situations or inefficient usage of space. For example, in crowded touristic cities such as Venice (Italy), crowd steering upon crowd forecasting can avoid preventing tourists to find difficulties in moving along Venice's narrow streets. In large airports, crowd forecasting can help directing people to different security gates before long queues appear.

## 2.5. Epidemiology

As of November 2020, the waves of the COVID-19 pandemic reached their highest peak in Europe, and the data collected from medical swabs is the first ground-truth dataset that we have about its size. However, there are studies that estimate the real numbers of the pandemics at least ten times larger than what reported from swabs data [31]. Moreover, there is a wide range of research showing that human mobility is the primary reason for the spread of the viruses [32–34]. In these conditions, monitoring and controlling human mobility and contacts means also monitoring and controlling the spread of COVID-19. This is particularly important for the stability of the national healthcare system of a country, of its hospitals and in particular of their intensive care units.

To get the situation under control, European governments have put in place a series of contact tracing techniques and measures including contact tracing apps. However, despite their potential, these tracing methods and apps have not worked so well in the US and in EU countries [35,36]. They seem to work better in Asian countries where cultural approach to these issues and privacy constraints are much more relaxed than in the EU [35]. As an alternative to contact tracing apps, researchers have pointed out the importance of mobile data (Call Description Records) for controlling the COVID-19 pandemics and evaluating the effectiveness of control measures such as physical distancing [37,38]. It is worth noticing however that the CDR data is currently scarcely used for tracking the COVID-19 spread. Although CDR data value for epidemiological modelling in general has been proven in several cases [39], privacy issues and in particular legislation hampers and limits their use.

## 3. Technologies and Data for Crowd Sensing

Different works on crowd sensing have shown that, in the IoT era, there are a multitude of data sources that can be successfully exploited in order to monitor crowds [40,41]. These different data sources are the main object of this section. We will go through the different data types explaining how they might help in studying crowd dynamics and, eventually, in producing forecasts.

## 3.1. Aggregated Call Description Records Data

Mobile phones are ubiquitous. No matter the level of economic development in a given country, its urban and rural areas will have full coverage of mobile phone networks reaching almost the entire population. Currently, the technology behind mobile communication is evolving from 4.5G to 5G which will make available massive bandwidths and increase the connection speed by a factor of 100 [42]. This means more data will be generated in

the same fraction of time. The data are normally collected by the mobile phone operators for business operations and technical purposes, offer an enormous opportunity to study human activity at a large scale. However, privacy concerns about CDR data release and in particular the possibility of tracking user trajectories in time and space, lead mobile network providers aggregating the data. The process of aggregation counts the number of mobile phones in a given area with a time sampling rate. This procedure allows preserving the privacy of users, as there is no way of singling out a particular mobile user. Since May 2018, all European countries have to comply with General Data Protection Regulation (GDPR) on this topic but there are countries such as China where CDR data that contain individual trajectories can be released.

Aggregated mobile phone data, which are shown in Figure 1, are arranged in a grid format where each cell of the grid contains several active mobile users in a given moment. These observations form a time series for each cell. Regarding the use and utility in studying crowd dynamics, aggregated CDR data is probably the best data source researchers could exploit. In fact, it allows for both real-time large scale crowd monitoring and crowd dynamics forecasting in the long term. There are countless studies which use sensing techniques from CDR data for studying crowd density and dynamics. The reader might consider the survey in [43] as a valid reference point.

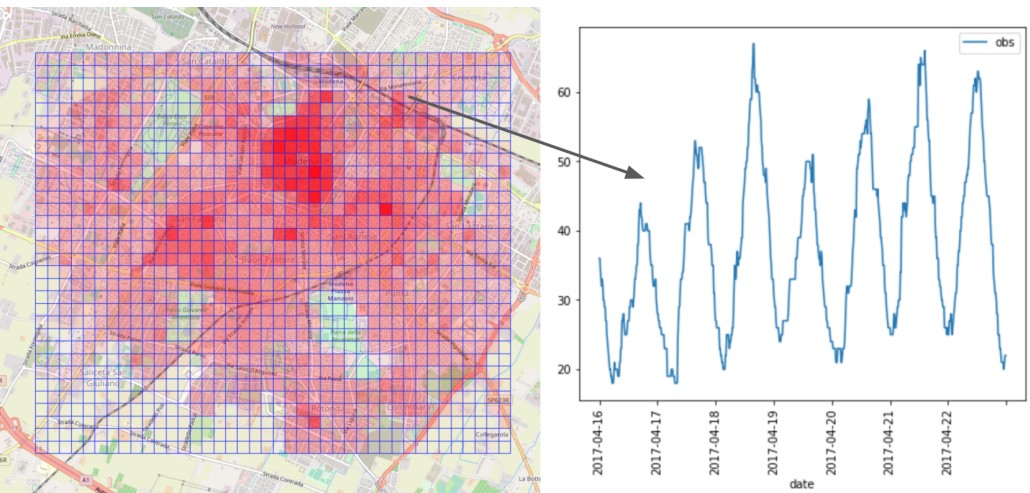

**Figure 1.** Aggregated mobile phone data over an urban area of Northern Italy. The geographic representation on the left shows a grid of aggregated data. Each cell of the grid contains a time series (shown on the right) of the number of people observed in that cell given a time sampling rate. Data source is one of the major Italian companies in telecommunications.

### 3.2. Wi-Fi Data

Another data source that allows collecting access logs from local networks is Wi-Fi, which has been shown to be highly reliable for monitoring crowds both indoors and outdoors [44–46]. The standard IEEE 802.11, commonly called Wi-Fi, is the world's most widely used wireless computer networking standard and is currently evolving to its latest version IEEE 802.11be (or Wi-Fi7) with even higher throughput [47]. Whether in homes and offices, or in public places and squares, Wi-Fi networks allow electronic devices such as smartphones to connect to the Internet reliably. As an example, Figure 2 shows the locations of several public access points in Emilia Romagna administrative region, Italy. (https://digitale.regione.emilia-romagna.it/ader-per-te/esplora/mappa-wifi)

However, using Wi-Fi data only to monitor and forecast the crowd size and dynamics may yield inaccurate results. For this reason, some works in the literature propose to integrate Wi-Fi data with other data sources, such as data coming from stereoscopic cameras [48] in order to estimate the crowd size, or from an automated people counting system [49] in order to better approximate crowd sizes.

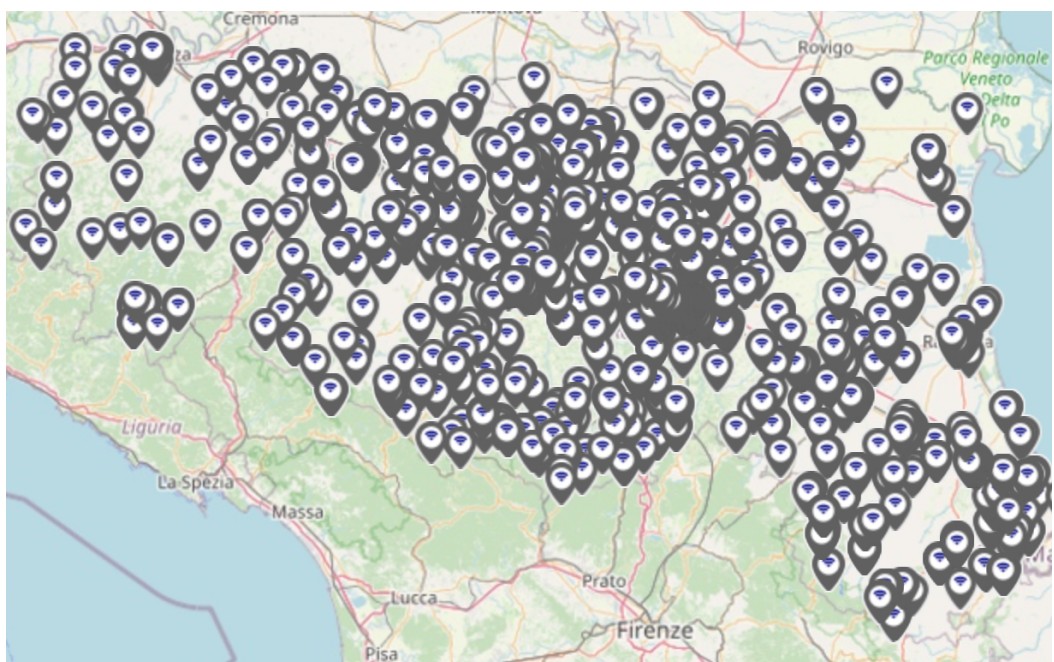

**Figure 2.** The map shows the locations of free Wi-Fi public access points in the Emilia Romagna administrative region, Italy.

*3.3. Social Media Data*

The GPS system and the related technology allows for countless applications [50]. At the same time, the huge growth of the number of users of social media platforms such as Twitter, and other location-based social media services, has led users to expose facts and opinions besides their current location. The resulting data are an interesting coupling of opinions and locations, thus transforming social media users in a sort of "social and physical sensors", continuously creating data about places and collective opinions [51]. Figure 3 is a heat map graphics built with Twitter spatio-temporal data that shows patterns of people tweeting during different moments of the day. The data (text and location) can be obtained by querying with specific time and location parameters the API services of the aforementioned social media platforms as, for example, Twitter. (https://developer.twitter.com/en/docs) Unfortunately, the filtering algorithms that social network platforms apply to those data greatly reduce the number of available data with these characteristics: in fact, only about 1% of the data is released with such features. Though limited and sparse in their size, the location data coming from social networks can still offer a good opportunity to study crowds as different studies show [52,53]. Finally, one of the best uses that researchers can make of location-based social media data is the use it with other location data, typically CDR data as in [22,54].

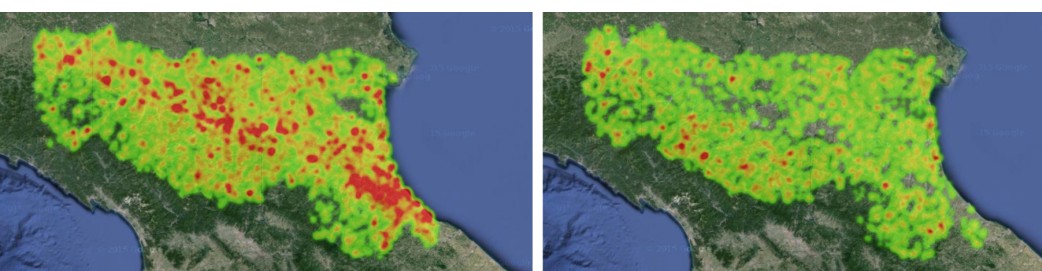

**Figure 3.** Heatmap obtained with spatio-temporal data from Twitter. The picture shows patterns that emerge during the different moments of the day. In particular, the graphics on the right shows a recurrent pattern that typically appears in the morning, while the one on the left is an evening pattern.

### 3.4. Cameras

An additional source of data that can be used in the context of mobility monitoring is given by cameras, such as those placed in urban areas. Computer vision approaches are typically used in order to process this kind of data and include the extracted information within forecasting models. Privacy issues clearly also have to be taken into account [55]. Many years of vision technology developments have gone past the conventional frame camera by producing different types of cameras such thermal cameras [56] and event cameras [57]. Figure 4 shows pictures taken from real-time camera view of Piazza San Marco (Venice) and Piazza Navona (Rome), two major Italian squares, which are accessible online (https://www.skylinewebcams.com/it/).

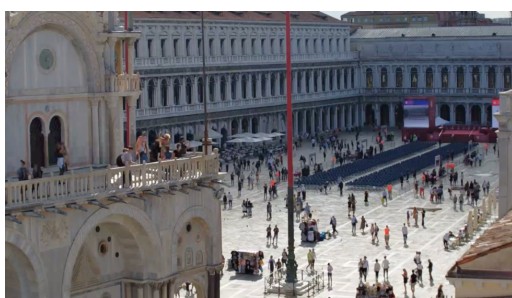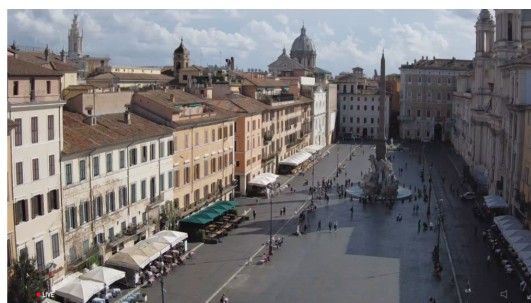

**Figure 4.** Public camera view of Piazza San Marco (Venice) and Piazza Navona (Rome) as seen from a real-time camera view online service.

### 3.5. Taxi Trajectories Data

Recent years (prior to the 2020 pandemics) have seen an increasing trend in demand for mobility services, and thus measuring and understanding traffic activity is now more important than ever. In this sense, taxi data trajectories are an important data source that represents an additional component for the improvement of mobility analysis and forecasting. There are different taxi service providers which release the data under an open-data philosophy. To protect privacy while allowing for aggregate analyses, such data consists of trip records which include features such as pick-up date, time, and taxi zone location. An example of such kind of taxi trajectories data is the New York City Taxi and Limousine Commission's (TLC) taxi service dataset, where each observation shows the number of NYC taxi passengers that are using the service in a given moment of the day with a sampling of 30 min (https://www1.nyc.gov/site/tlc/about/tlc-trip-record-data.page). As an example of such data, the time series depicted in Figure 5 shows how the number of passengers evolves over time.

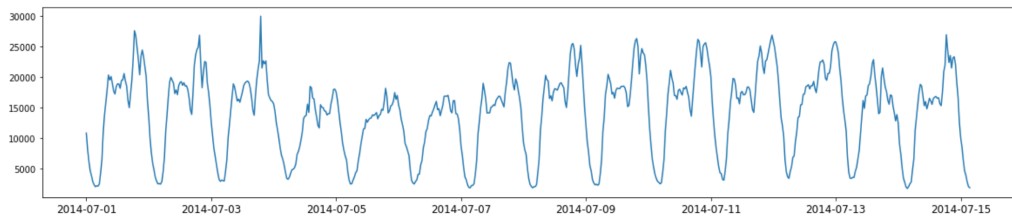

**Figure 5.** The time series shows the number of NYC taxi passengers that are using the service in a given moment during a period of 15 days.

Some recent and relevant works exploiting taxi trajectories for the study of mobility dynamics can be found in [58,59].

## 4. Approaches to Crowd Modeling and Mobility Forecasting

From the description of the variety of data that can be used to perform crowd modeling and mobility forecasting, it is quite obvious that the design of suitable methodologies strongly depends on the data that are available in each given scenario. In this

section we will now overview the main techniques that have been developed in the most common situations.

### 4.1. Statistical Approaches to Time-Series Forecasting

Statistical approaches have historically been used to perform time-series forecasting in several areas. The majority of approaches exploit linear models: this is the case of auto-regressive models, as well as of models that combine an auto-regressive component with a moving average, such as the well-known ARIMA model, that was generalized also to seasonal patterns and multivariate series. Besides being very effective in many forecasting scenarios, these methods are also interpretable, which is an interesting features when aiming at producing explainable predictions. ARIMA has been used for crowd counts forecasting [20], to perform forecasts on international tourism time series data [60], and to correlate traffic congestions to socio-economic indicators [61].

Another statistical method used for forecasting crowd counts is also generalized autoregressive conditional heteroskedastic (GARCH) model. For example, the authors in [62] use GARCH models to compute prediction intervals associated with the forecasts obtained with ARIMA on Wi-Fi data.

Several mathematical models have also been proposed to characterize human mobility dynamics, ranging from a micro-scale, typically with a low density, up to a macro-scale, typically with a high density. Such approaches include cellular automata, social force models, velocity-based models, continuum models, hybrid models, behavioral models and network models [63,64].

Nevertheless, these statistical approaches are now being overcome by more recent approaches based on deep learning architectures, which will be described in the next subsection. For a detailed experimental comparison between statistical and deep learning approaches for crowd mobility forecasting, the reader may also refer to recent experimental surveys [65,66].

### 4.2. Deep Learning for Time-Series Forecasting

In recent years, the literature in the field of crowd forecasting has been dominated by methods that exploit deep learning approaches. In fact, deep neural networks have shown remarkable performance in many forecasting scenarios, including the domain of mobility and telecommunications [67,68].

Recurrent neural networks such as Long Short-Term Memory (LSTM) networks and convolutional neural networks (CNN) with one-dimensional filters are typically employed in the case of univariate time-series [20,69]. CNNs have been originally designed for the processing of images, but they have been further generalized to handle other kinds of signals [70,71]. By using temporal receptive fields, CNNs applied to signals learn filters that extract features from raw data. RNNs are instead neural networks that can naturally process sequences, and are therefore widely applied to the domain of time-series forecasting. A special category of RNNs, named Long Short Term Memory (LSTM) networks, are designed so as to capture long-range dependencies, and thus they have been particularly successful in those scenarios where this is an important feature.

Starting from these basic models, presently the majority of methods based on deep learning deal with the development of spatio-temporal architectures capable of capturing dependencies among different time-series, that typically correspond to different cells in the considered geographical area.

A huge amount of works exist in this vein, that typically consider a city-wide scenario. Many different spatio-temporal neural networks have been designed to tackle this problem [24,72–74]. An integration of a similar neural architecture with geo-referenced Twitter data was proposed in [75]. Spatial-Temporal Densely Connected Networks have also been proposed to predict both inflow and outflow of crowds in every region of a city [76] Similarly, a tailored architecture, named expand-and-squeeze network, has been introduced in [77] for crowd flow prediction in a metropolis. Graph convolutional neural networks,

widely employed in many machine learning tasks, have also been considered to capture spatial dependencies [78]. In other cases, background knowledge of the spatio-temporal dependencies between different areas can be exploited to design tailored models [79]. To improve the interpretability of deep learning approaches–a major issue in artificial intelligence presently–the mechanism of attention can be exploited so as to identify the most valuable portions of the data, and of the model, for the task at hand [80]. Attentive models have also been proposed to predict crowd flow in residual networks [81] as well as in densely connected convolutional networks and long short-term memory networks [82]. The problem has also been formulated as a sequence-to-sequence prediction, by exploiting generative adversarial networks to perform multi-step spatio-temporal crowd flow prediction at a city-wide level [83]. Finally, an original approach is given by the work in [84] which proposes a frequency-aware spatio-temporal convolutional neural network. Such network has various kernels of different sizes which can model the temporal correlations with multi-scale frequencies.

### 4.3. Approaches Based on Computer Vision

When the input data consists of images and videos collected from cameras, computer vision approaches are to be exploited. Some early applications of such methods for crowd analysis were counting the number of passing people. Safety issues for crowd management can be found respectively in [85,86]. A more recent work combines Wi-Fi data captured from mobile devices and stereoscopic cameras to build a model for people counting [48].

In the last ten years, CNNs have become the standard methodology in order to detect, analyze and eventually predict crowds and their dynamics in pictures and videos. Therefore, from counting the number of passing people, the application scenario has evolved to crowd segmentation [87] and crowd counting [88,89]. Finally, some interesting approaches to problems such as detecting and forecasting crowd density based on CNNs are given in [90,91]. CNNs have also been used to perform crowd motion analysis in large events, to process streaming data from cameras [92]. Similar techniques have been employed also to perform behavior recognition in crowds [93] and congestion level analysis [94]. Even in this case, the reader may refer to a dedicated survey for a thorough analysis of related approaches [95].

### 4.4. Methods for Motion and Trajectory Estimation

A large body of works exist that leverage the power of a computer vision technique called optical flow modeling in order to estimate and predict crowd motion. For example, in [96] crowd prediction is delivered in the form of a series of predictive maps based on an optical flow estimation algorithm. The series of such maps represents the crowd distribution at different points in time, as predicted by the optical flow estimator. To the same category of works belongs the paper in [97], which proposes a method for the detection of anomalous events in crowds, by exploiting histograms of optical flow orientation and magnitude, with the additional challenge given by the unpredictable nature of such events. Further interesting works on crowd analysis and crowd flow prediction using optical flow can be found in [98,99].

Another category of methods is very well represented by the work in [100], which aims to predict crowd movement from crowd trajectories using agent-based motion models (AMMs). This method applies several AMMs, also called exemplar-AMMs, to describe crowd movements. The authors then propose an optimization framework that measures the similarity of crowd trajectories with those of exemplar-AMMs in order to produce a crowd motion prediction feature. In a similar way, the work in [101] proposes a modeling framework to construct agent-based crowd models from real-world video data. The crowd model obtained can then generate crowd behaviors that match those observed in the video. This match and can be used to predict trajectories of pedestrians in the same scenario.

On a different perspective, another wide spectrum of works have been devoted to the design of models to simulate and forecast crowd motion [64]. One way perform

analysis and modeling of crowd trajectories is by exploiting hidden Markov models [102], hidden semi-Markov models [103], as well as mixtures of multiple random Markov chains combined with clustering techniques [104].

Finally, an interesting research stream is given by works that use deep neural networks in order to recognize and thus predict crowd movements and flow. For example, a very recent work [105] addresses the challenge of predicting the crowd flows in especially irregular regions of the city area. The authors use a CNN which has been extended to handle the spatial information as a graph. This way they build a multi-view graph convolutional network in order to deliver forecasts for the crowd flow in irregular areas of the city. The authors of [106] instead, present a deep learning framework that aims to predict the long-term flow of crowds in realistic environments. Their approach is based on a novel deep neural network which encodes crowd scenarios into compact, fixed-size representations that losslessly represent the environment.

*4.5. Combined Approaches*

In this subsection, we list a series of approaches on crowd flow prediction that exploit and combine more than just one data source or more than one prediction method. By combined approach, we thus mean those approaches that deliver prediction in such ways that the final results can have a greater value than simply predicting with just one data source or method. For example, the study in [107] presents novel algorithms to fuse data collected by wireless sensors with additional data sources from Bluetooth sensors in order to improve the estimation of pedestrian flows and counting results. For the same purpose, i.e., estimating and counting people in crowds, although using different types of data, the work in [54] presents an approach for proving the relationship between the number of people in restricted areas such as airports and stadiums and the mobile activity recorded by the providers and the online service Twitter.

Even the work presented in [108] combines different data sources such as weather data and historical data of people presence at sub-way stations. In particular, this study combines time-variant features such as weather conditions or historical traffic at subway stations, with data of station profile in order to deliver accurate predictions of crowd flow. On the same topic, that of metro station passengers flow prediction, the work in [109] proposes a novel deep learning framework for spatio-temporal correlation called STP-TrellisNet. The framework can capture the short- and long-term temporal correlations with graph convolution and has the ability to capture dynamic graph-structured correlations. The authors of this paper, claim that their framework can outperform the state-of-the-art baselines. In a similar way, the study in [110] presents and approach based on big data in order to forecast citywide crowd flows. In particular it combines different data sources such as human mobility data, weather conditions, and road network data. By decomposing the patterns in the data into three components: periodic, trend and residual, the authors state that they are able to model the multiple complex factors affecting crowd flows in a much better way.

However, crowd flow modelling and prediction can be of great value also for monitoring and eventually better control the way viruses spread in urban areas. The work in [111], for example, combines anonymized mobile device locations with flight data in order to track COVID-19 spread.

The main benefit of hybrid approaches in forecasting crowd flows is that they can combine not only different data sources but also different methods. For example in the work presented in [110], spatial and temporal dependencies have been modeled with Markov Random Fields and Bayesian networks for crowd flow forecasting. In particular, a Gaussian Markov random fields model that can cope with noisy and missing data, and a residual model which exploits the spatio-temporal dependence among different flows and regions of the city, as well as the effect of weather. A combination of predicting methods is also given in [112], where authors combine a spatio-temporal residual network (ST-ResNet) with a long short-term memory (LSTM) neural network.

## 5. Open Challenges

In this section, we discuss the main open challenges for the further development of the field: (i) balancing data availability and privacy, (ii) dealing with scenarios where few data sources are available via transfer learning, (iii) data fusion and integration to analyze patterns from multiple facets.

### 5.1. Privacy Issues

Among all the sensing technologies, probably the one that can harm privacy the most is video-camera technology with its capability of associating a face to a name univocally. Yet, other sensing technologies such as mobile phones or Wi-Fi trajectories have shown to be non the less harmful to an individual's privacy [113]. In addition to that, by joining in space and time different types of datasets, the risks can grow even more, since anonymized data can be quite easily de-anonymized [114]. Within this context, it is clear that data processing techniques are required in order to protect data privacy. When it comes to privacy in mobility data, the question is always the same: how can we aggregate or perturbate data in such a way that individual data privacy is preserved but at the same time the utility of data is not destroyed? A trade-off between data privacy and utility has to be considered [115].

When extending this reasoning towards crowd data then the question becomes how we can get more from data, more details (e.g., type of people in the crowd, age, etc.) without undermining privacy. It is not easy to find in the existing literature works that have already attempted to answer this question. The work in [116] presents a privacy-preserving crowd monitoring system but only for counting the number of people in the crowd without extracting any relevant features. The work presented in [117] presents a data-driven crowd understanding approach to estimate different crowd behavior features such as collectiveness and cohesiveness on data collected from video-cameras, yet without considering privacy issues. In general, the promise for characterizing crowds using anagraphic features such as age, for example, remains unfulfilled. Other works instead deal with the problem of privacy but without characterizing the crowd such as, for example [118].

### 5.2. Transfer Learning

Despite the availability of almost any kind of data that we have presently, there can still be situations where data scarcity could be a problem. This happens, for example, when a relatively new urban area/city with little historical data is considered. In such cases, it is hard to build accurate models or to leverage large collections of historical data for learning, but we can instead use what has been learnt in similar situations with data from other cities.

There are different works in the literature that attempt to answer this question through the use of transfer learning [119]. For example the works in [120,121] consider crowd flow forecasting and electric load forecasting problems respectively. They learn representations such as crowd flows and electric loads in data-rich cities and then transfer this knowledge to target cities in order to make predictions. This type of transfer learning is called cross-region or cross-city transfer learning. Although the idea is quite intuitive, the practical implementation is full of difficulties of different nature. For example, different urban areas have different development levels, which makes a direct transfer usually useless [122]. Thus, the results of a transfer learning process need careful validation and seem to be application-dependent, i.e., not easily generalizable for other applications.

### 5.3. Data Fusion and Integration

One of the main issue with the majority of current works is that most research is conducted in isolation as individual data mining "exercises". As the number of available data sources and algorithms to process them are constantly increasing, the perception is that the algorithms to extract relevant information from data are already there. The important challenge is to find ways of combining them together so that results coming from one dataset can validate and further describe results from other data. The current state of the art

in performing this integration is associated with ensemble-classifiers and ontology-driven integration. Unfortunately, ensemble-classifiers tend to be rather inflexible in that they require a certain level of homogeneity among classifiers to be merged, ontology-driven integration works well in small scale scenarios, but further research is needed to apply it to large-scale mobility [123]. The survey papers in [124,125] can give to the reader a better understanding of urban data fusion and in particular of the fusion methods based on deep learning.

## 6. Conclusions

Sensing and predicting crowd movements in smart cities is of fundamental importance. Such a task can be crucial for the improvement of urban planning, mobility and safety and, as such, it involves many challenges from the point of view of data collection, processing, analysis, and also privacy.

In this paper, we provided an overview of several aspects of this important problem. We first presented the potential applications where accurate crowd sensing and forecasting could play a crucial role. Then, we described the data and the methodological approaches that are currently exploited to sense crowds within an urban environment, and to perform forecasts about future behavior. We finally pointed out future research challenges, focusing on the problem of data fusion across different sources, on the transfer of knowledge across different tasks, and, last but not least, on the privacy issues that arise when dealing with mobility data.

We do believe that this research area represents a crucial topic for the development of our cities and of our society.

**Author Contributions:** A.C. investigation, writing—review and editing, M.L. investigation, writing—review and editing, M.M. investigation, project administration, funding acquisition, F.Z. conceptualization, review and editing, project administration, funding acquisition. All authors have read and agreed to the published version of the manuscript.

**Funding:** Work supported by: the POR-FESR 2014-2020 Project: POLIcy Support systEm for smart citY data governancE - POLIS-EYE. PG/2018/631990. CUP E21F18000200007; and the MIUR PRIN 2017 Project: Fluidware.

**Institutional Review Board Statement:** Not applicable.

**Informed Consent Statement:** Not applicable.

**Data Availability Statement:** We have shared the data by indicating with a link each dataset source.

**Conflicts of Interest:** There are no conflicts of interest between the authors and the funding parties.

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
