# Peer review of "Sensing and Forecasting Crowd Distribution in Smart Cities: Potentials and Approaches"

_2624-831X, doi:10.3390/iot2010003_

Round 1
Reviewer 1 Report
This paper investigates traditional and recent works of the topic of sensing and crowd distribution in cities.
The authors’ work, more of an extensive literature review, sheds some light on the overall development in the topic, in an organized structure and a comprehensive way. It does provide a brief guidance to the new-comers of this field.
However, there are several problems in the paper, especially in breadth and depth, which make the paper cannot be fully appreciated. The shortcomings are identified as follows.
- In Section 3, the right sub-figure of Figure 1 is not self-evident enough, with missing axis labels and explanations. Meanwhile, the data source is not explicitly pointed out.
- In Section 3.2, the purpose of Figure 2 in the context is ambiguous. It is stated only by that “as an example, Figure 2 shows the locations of a number of public access points in Rimini, Italy.” However, no clear high/low density or bias is depicted in the figure, which makes the aim really confusing.
- In Section 3, the authors do not consider some commonly-used and privacy-insensitive data sources such as taxi trajectories, taxi orders, etc., which can be leveraged in city sensing.
- In Section 4.1, the statistical approaches are poorly elaborated. Merely ARMIA is introduced. However, some other important and effective ones based on mathematical-related methods, e.g. optimization formulation and derivations, are omitted.
- The authors fail to properly cite several past literature (e.g., [1-3]) highly related to this work.
[1] Predicting Citywide Crowd Flows in Irregular Regions Using Multi-View Graph Convolutional Networks, TKDE 2020.
[2] STP-TrellisNets: Spatial-Temporal Parallel TrellisNets for Metro Station Passenger Flow Prediction, CIKM 2020.
[3] A Frequency-Aware Spatio-Temporal Network for Traffic Flow Prediction, DASFAA 2019.
- In Section 4.5, the claimed concept of “hybrid” is misleading. For instance, the given example of fusing wireless sensors data with additional Bluetooth data, to a certain extent, is more of one of the detailed design tricks used in various approaches, such as RNN, instead of a brand-new “approach” as suggested in the paragraphing.
Author Response
Dear reviewer,
please find below our answers.
1- We have changed Figure 1 in order to better represent the geographic location and show that the grid is over a geographic map. We better explained this point in the caption. Regarding the data source, we are not allowed to explicitly share the data source: we can only state that the source is one of the major Italian companies in telecommunications, which we added to the caption.
2- The purpose of Figure 2 is to show that Wi-Fi access points are now ubiquitous and provide open access to everyone. However, we have now changed the picture in order to show the whole Emilia Romagna administrative region.
3- We have now inserted this category of data in subsection 3.5 and considered different relevant works using taxi trajectories data. We want to thank the reviewer for the suggestion.
4- Besides ARIMA we have now also added a paper which uses GARCH models as a way to predict crowd counts from Wi-Fi data. We also discussed mathematical models that extend micro-scale (low-density) models up to a macro-scale (with high-density).
5- The work in [1] was already on our list of references. We have now also added the suggested references [2] and [3], for which we thank the reviewer.
6- We have now changed the term “hybrid” in subsection 4.5 with the term “combined” in order to indicate crowd prediction approaches that combine more than two different types of data or prediction methods.
Reviewer 2 Report
This seems to be a "survey paper", as it performs a literature survey and provides the facts in a classified ways, such as (a) Potentials of Crowd Sensing, (b) Technologies for Crowd Sensing, (c) Approaches for Crowed Sending and Modeling, and (d) Challenges of Crowed Sensing. Thus there is no originality, and no unique work is reported by the authors. The abstract should mention that it is a survey paper.
There are several conflicting statements: e.g. abstract says "sensing and predicting the movements of crowds in modern cities is of
fundamental importance", whereas conclusion says, "Sensing and predicting crowd movements has nowadays become a fundamental problem". I think abstract is correct, because there are lot new innovations taking place around crowed sensing.
The "Open Challenges" section is vague. Some of the challenges mentioned are directly related to the crowd sensing, but some are not. For example, Data Fusion and Integration (researches conducted in isolation) is not crowed sensing challenge, it is an operational issue related to streamlining the research done in different countries. Similarly, data validation is not an issue of Crowed Sensing.
The reviewer could not find any new information in the paper.
Author Response
Dear reviewer,
please find below our answers.
1- We have now clarified the survey nature of our paper directly in the abstract.
2- We have now corrected the conflicting statement by aligning our "Conclusions" with what we stated in the abstract.
3- We have added some more relevant works about data fusion in order to give the reader a better understanding.
Round 2
Reviewer 1 Report
The authors have addressed my comments to the previous version. I do not have any further comments, and recommend that this paper be accepted.
Author Response
We want to thank the reviewer for his comments and suggestions.
Reviewer 2 Report
The author has addressed the comments.
Though the novelty and scientific soundness are below average, the paper presents some good applications, future research challenges, the some privacy issues of mobility data.
Author Response
We thank the reviewer for his comments and suggestions . We believe that our survey article can be helpful to interested researchers and readers in understanding the actual state of the field better.